# Informed Temporal Modeling via Logical Specification of Factorial LSTMs

## Abstract

Consider a world in which events occur that involve various entities. Learning how to predict future events from patterns of past events becomes more difficult as we consider more types of events. Many of the patterns detected in the dataset by an ordinary LSTM will be spurious since the number of potential pairwise correlations, for example, grows quadratically with the number of events. We propose a type of *factorial* LSTM architecture where different blocks of LSTM cells are responsible for capturing different aspects of the world state. We use Datalog rules to specify how to derive the LSTM structure from a database of facts about the entities in the world. This is analogous to how a probabilistic relational model (Getoor & Taskar, 2007) specifies a recipe for deriving a graphical model structure from a database. In both cases, the goal is to obtain useful inductive biases by encoding informed independence assumptions into the model. We specifically consider the neural Hawkes process, which uses an LSTM to modulate the rate of instantaneous events in continuous time. In both synthetic and real-world domains, we show that we obtain better generalization by using appropriate factorial designs specified by simple Datalog programs.

## 1 Introduction

Temporal sequence data is abundant in applied machine learning. A common task is to impute missing events, e.g., to predict the future from the past. Often this is done by fitting a generative probability model. For evenly spaced sequences, historically popular models have included hidden Markov models and discrete-time linear dynamical systems, with more recent interest in recurrent neural network models such as LSTMs. For irregularly spaced sequences, a good starting point is the Hawkes process, a self-exciting temporal point process; many variations and enhancements have been published, including neural variants using LSTMs.

All of these models can be described schematically by Figure 1a. Events $e_i, e_{i+1}, \dots$ are assumed to be conditionally independent of previous events, given the system state $\mathbf{s}_i$ (which may or may not be fully known given events $e_1, \dots, e_i$). That is, $\mathbf{s}_i$ is enough to determine the joint distribution of the $i^{\text{th}}$ event and the updated state $\mathbf{s}_{i+1}$, which is needed to recursively predict all subsequent events.

Figure 1a and its caption show the three types of influence in the model. The `update`, `affect`, and `depend` arrows are characterized by parameters of the model. In the case of a recurrent neural network, these are the transition, input, and output matrices.

Our main idea in this paper is to inject structural zeros into these weight matrices. Structural zeros are weights that are fixed at zero regardless of the model parameters. In other words, we will remove many connections (synapses) from both the recurrent and non-recurrent portions of the neural network. Parameter estimation must use the sparse remaining connections to explain the observed data.

Specifically, we partition the neural state $\mathbf{s}_i \in \mathbb{R}^d$ into a number of node blocks. Different node blocks are intended to capture different aspects of the world's state at step $i$. By zeroing out rectangular blocks of the weight matrix, we will restrict how these node blocks interact with the events and with one another. An example is depicted in Figures 1b (`affect`, `depend`) and 1d (`update`).

In addition, by reusing nonzero blocks within a weight matrix, we can stipulate (for example) that event $e$ **affect**s node block $b$ in the same way in which event $e'$ **affect**s node block $b'$. Such parameter tying makes it possible to generalize from frequent events to rare events of the same type.

Although our present experiments are small, we are motivated by the challenges of scale. Real-world domains may have millions of event types, including many rare types. To model organizational behavior, we might consider a dataset of meetings and emails in a large organization. To model supply chains, we might consider purchases of goods and services around the world. In an unrestricted model, anything in the past could potentially influence anything in the future, making estimation extremely difficult. Structural zeroes and parameter tying, if chosen carefully, should help us avoid overfitting to coincidental patterns in the data.

Analogous architectures have been proposed in the world of graphical models and causal models. Indeed, to write down such a model is to explicitly allow specific direct interactions and forbid the rest. For example, the edges of a Gaussian graphical model explicitly indicate which blocks of the inverse covariance matrix are allowed to be nonzero. Some such models reuse blocks (Hojsgaard & Lauritzen, 2008). As another example, a factorial HMM (Ghahramani & Jordan, 1997)—an HMM whose states are $m$-tuples—can be regarded as a simple example of our architecture. The state $\mathbf{s}_i$ can be represented using $m$ node blocks, each of which is a 1-hot vector that encodes the value of a different tuple element. The key aspect of a factorial HMM is that the stochastic transition matrix (**update** in Figure 1d) is fully block-diagonal. The **affect** matrix is 0, since the HMM graphical model does not feed the output back into the next state; the **depend** matrix is unrestricted.

But how do we know which interactions to allow and which to forbid? This is a domain-specific modeling question. In general, we would like to exploit the observation that events are structured objects with participants (which is why the number of possible event types is often large). For example, a travel event involves both a person and a place. We might assume that the probability that Alice travels to Chicago depends only on Alice's state, the states of Alice's family members, and even the state of affairs in Chicago. Given that modeling assumption, parameter estimation cannot try to derive this probability (presumably incorrectly) from the state of the coal market.

These kinds of systematic dependencies can be elegantly written down using Datalog rules, as we will show. Datalog rules can refer to database facts, such as the fact that Alice is a person and that she is related to other people. Given these facts, we use Datalog rules to automatically generate the set of possible events and node blocks, and the ways in which they influence one another. Datalog makes it easy to give structured names to the events and node blocks. The rules can inspect these structures via pattern-matching.

In short, our contribution is to show how to use a Datalog program to systematically derive a constrained neural architecture from a database. Datalog is a blend of logic and databases, both of which have previously been used in various formalisms for deriving a graphical model architecture from a database (Getoor & Taskar, 2007).

## 2 PRELIMINARIES[1]

Our methods could be applied to RNN sequence models. In this setting, each possible event type would derive its *unnormalized* probability from selected node blocks of state $\mathbf{s}_i$. Normalizing these probabilities to sum to 1 would yield the model's distribution for event $e_i$. Only the normalizing constant would depend on all node blocks.

In this paper, we focus on the even more natural setting of *real-time* events. Here no normalizing constant is needed: the events are not in competition. As we will see in section 5.1, it is now even possible for different node blocks to generate completely independent sequences of timestamped events. The observed dataset is formed by taking the union of these sequences.

In the real-time setting, event $e_i$ has the form $k_i@t_i$ where $k_i \in \mathcal{K}$ is the *type* of the event and $t_i \in \mathbb{R}$ is its *time*. The probability of an event of type $k$ at any specific instant $t$ is infinitesimal. We will model how this infinitesimal probability **depend**s on selected node blocks of $\mathbf{s}_i$. There is no danger that two events will ever occur at the same instant, i.e., the probability of this is 0.

---

[1]Our conventions of mathematical notation mainly follow those given by Mei & Eisner (2017, section 2).

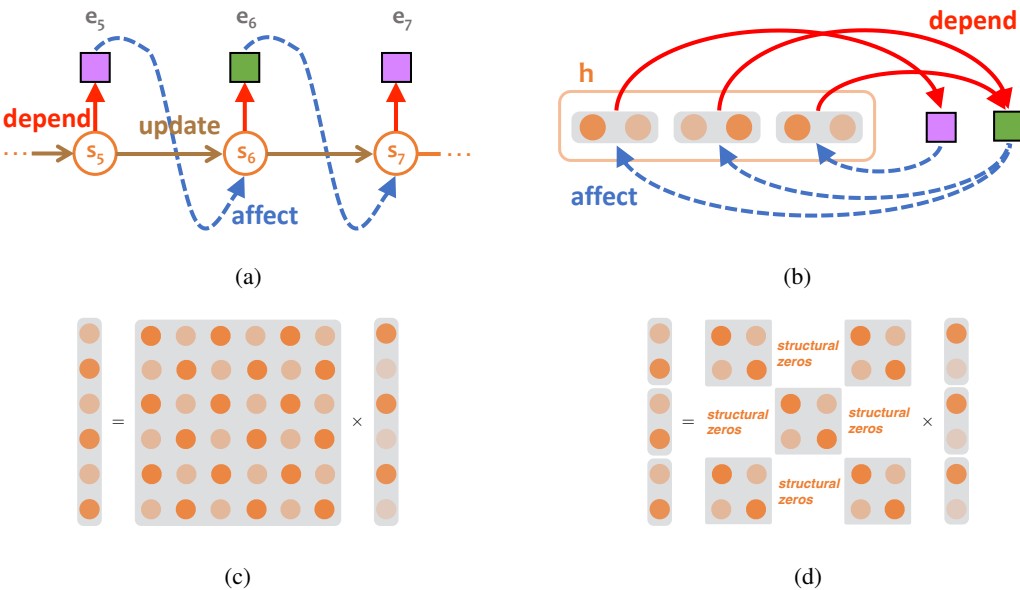

Figure 1: Diagram (a) shows how the system state $\mathbf{s}_i$ evolves over time, generating observable events along the way. The event $e_i$ **depend**s stochastically on the system's current state $\mathbf{s}_i$. The current state then **update**s either deterministically or stochastically to $\mathbf{s}_{i+1}$, where $e_i$ also **affect**s this new state. Diagram (b) shows how the state may be factored into 3 node blocks. In this example, the relative probability of green events now **depend**s on only the rightmost ⅔ of the state, and if a green event occurs, it **affect**s only the leftmost ⅔ of the state. The **update** computation involves a matrix multiplication shown in (c); by zeroing out some blocks of the matrix, as illustrated in (d), we can ensure that the **update**d center node block is not affected by the preceding state of the other node blocks at time $i$, nor vice-versa.

We begin by describing our baseline model for this setting, drawn from Mei & Eisner (2017).

## 2.1 BASELINE MODEL: THE NEURAL HAWKES PROCESS

In general, a multivariate point process is a distribution over possible sequences of events $e_1 = k_1@t_1, e_2 = k_2@t_2, \ldots$ where $0 < t_1 < t_2 < \ldots$. A common paradigm for defining such processes, starting with Hawkes (1971), is to describe their temporal evolution as in Figure 1a. Each $\mathbf{s}_i$ is deterministically computed from $\mathbf{s}_{i-1}$ (**update**) and $e_{i-1}$ (**affect**), according to some formula, so by induction, $\mathbf{s}_i$ is a deterministic summary of the first $i-1$ events. $e_i = k_i@t_i$ is then emitted stochastically from some distribution parameterized by $\mathbf{s}_i$ (**depend**).

The structure of the **depend** distribution is the interesting part. $\mathbf{s}_i$ is used, for each event type $k \in \mathcal{K}$, to define some time-varying intensity function $\lambda_k : (t_{i-1}, \infty) \to \mathbb{R}_{\geq 0}$. This intensity function is treated as the parameter of an inhomogeneous Poisson process, which stochastically generates a set of future events of type $k$ at various times in $(t_{i-1}, \infty)$.[2] Thus, all these $|\mathcal{K}|$ Poisson processes together give us many events of the form $e = k@t$. The *first* such event—the one with the earliest time $t$—is taken to be the next event $e_i$. The remaining events are discarded (or in practice, never generated).

As our baseline method, we take the neural Hawkes process (Mei & Eisner, 2017) to be our method for computing $\mathbf{s}_i$ and defining the intensity function $\lambda_k$ from it. In that work, $\mathbf{s}_i$ actually describes a parametric function of the form $\mathbf{h} : (t_{i-1}, \infty) \to \mathbb{R}^d$, which describes how the hidden state of the system evolves following event $e_{i-1}$. That function is used to define the intensity functions via

$$\lambda_k(t) = f_k(\mathbf{v}_k^\top \mathbf{h}(t)) > 0 \tag{1}$$

---

[2]Under an inhomogenous Poisson process, disjoint intervals generate events independently, and the number of events on the interval $(a, b]$ is Poisson-distributed with mean $\int_a^b \lambda_k(t)\, dt$. Thus, on a sufficiently narrow interval $(t, t + dt]$, the probability of a single event is approximately $\lambda_k(t)\, dt$ and the probability of more than one event is approximately 0, with an error of $O(dt^2)$ in both cases.

so the parameters of **depend** are the vectors $\mathbf{v}_k$ and the monotonic functions $f_k$. Once $e_i = k_i@t_i$ has been sampled, the parameters for $\mathbf{s}_{i+1}$ are obtained by

$$\mathbf{s}_{i+1} \leftarrow \Psi(\mathbf{U}\mathbf{w}_{k_i} + \mathbf{V}\mathbf{h}(t_i)) \tag{2}$$

where $\Psi$ is inspired by the structure of an LSTM, the **affect** parameters are given by matrix $\mathbf{U}$ and the event embeddings $\mathbf{w}_k$, and the **depend** parameters are given by matrix $\mathbf{V}$.

In this paper, we will show an advantage to introducing structural zeroes into $\mathbf{v}_k$, $\mathbf{U}$, and $\mathbf{V}$.

## 2.2 STRUCTURED EVENTS AND NODE BLOCKS

In real world, atomic events typically involve a predicate and a few arguments (called entities in the following), in which case it makes sense to decompose an event type into a structured form[3] such as `email(alice,bob)`, `travel(bob,chicago)`, etc. For generality, we also allow entities to have structured forms when necessary.

Then naturally, in such a world with many entities, we would like to partition the state vector $\mathbf{h}(t)$ into a set of **node blocks** $\{\mathbf{h}_b(t)\}_{b \in \mathcal{B}}$ and associate node blocks with entities. For example, we may associate $\mathbf{h}_{\texttt{mind(alice)}}(t)$ to `alice` and $\mathbf{h}_{\texttt{mind(bob)}}(t)$ to `bob`. Note that `mind(alice)` is just an example of the kind of node blocks that can be associated with `alice`. There can be another node block associated with the physical condition of `alice` and be called `body(alice)`. Of course when there is only one node block associated with `alice`, we can also simply call it `alice`.

From now on, we use teal-colored typewriter font for events and orange-colored font for node blocks.

From Figure 1b, we already see that an event may only depend on and affect a subset of hidden nodes in $\mathbf{h}(t)$, and this further prompts us to figure out a way to describe our inductive biases on which node blocks are to determine the intensity of a given event as well as which node blocks are to be updated by one.

## 3 THE DATALOG INTERFACE

We propose a general interface based on Datalog—a declarative logic programming language—to assert our inductive biases into a deductive database as facts and rules. Then as each event happens, we can *query* the database to figure out which node blocks determine its intensity and which node blocks will be updated by it.

In this section, we walk through our Datalog interface by introducing its keywords one step a time. We write keywords in boldfaced typewriter font, and color-code them for both presentation and reading convenience. The colors we use are consistent with the examples in Figure 1.

### 3.1 THE `is_block` AND `is_event` KEYWORDS

We first need to specify what is a legal node block in our system by using the keyword `is_block`:

$$\texttt{is\_block}(b). \tag{3}$$

where $b$ can be replaced with a node block name like `alice`, `bob`, `chicago` and etc. Such a Datalog statement is a database fact.

Then we use the keyword `is_event` to specify what is a legal event type in our system:

$$\texttt{is\_event}(k). \tag{4}$$

where $k$ can be replaced with `email(alice,bob)`, `email(bob,alice)`, `travel(bob,chicago)` and etc. As we may have noticed, there may be many variants of `email(S,R)` where the variables `S` and `R` can take values as `alice`, `bob` and etc. To avoid writing a separate fact for each pair of `S` and `R`, we may summarize facts of the same pattern as a rule:

$$\underbrace{\texttt{is\_event(email(S,R))}}_{\text{head of rule}}\texttt{:-}\underbrace{\texttt{is\_block(S),is\_block(R).}}_{\text{body of rule}} \tag{5a}$$

---

[3]Similar structured representation of events has been common in natural language semantics (Davidson, 1967) and philosophy (Kim, 1993).

where `:-` is used to separate the head and body. Capitalized identifiers such as `S` and `R` denote variables. A rule mean: for any value of the variables, the head is known to be true if the body is known to be true. A fact such as `is_event(email(alice,bob))` is simply a rule with no body (so the `:-` is omitted), meaning that the body is vacuously true.

To figure out what event types are legal in our system, we can query the database by:

$$\text{is\_event}(\text{K})? \tag{6}$$

which returns *every* event type $k$ that instantiates `is_event`$(k)$. Note that, unlike a fact or rule that ends with a period (`.`), a query ends with a question mark (`?`).

## 3.2 THE `depend` KEYWORD

We can declare database rules and facts about which events depend on which node blocks using the `depend` keyword as:

$$\text{depend}(k,b)\text{:- } \text{condition}_1,\ldots,\text{condition}_N. \tag{7}$$

where $k$ and $b$ are replaced with Datalog variables or values for event and node block respectively, and `condition`$_1$`,...,condition`$_N$ stands for the body of rule. An example is as follows:

$$\text{depend}(\text{travel}(\text{bob},\text{chicago}), \text{ X})\text{:- } \text{resort}(\text{X}),\text{at}(\text{X},\text{chicago}). \tag{8a}$$

$$\text{depend}(\text{travel}(\text{bob},\text{chicago}), \text{ X})\text{:- } \text{friend}(\text{bob},\text{X}),\text{at}(\text{X},\text{chicago}). \tag{8b}$$

By querying the database for a given $k$ using

$$\text{depend}(k,\text{B})? \tag{9}$$

we get $\mathcal{B}_k^{\text{d}}$ that is the set of all the node blocks $b$ that instantiates `depend`$(k,b)$ and has superscript d for `depend`. Then we have:

$$\lambda_k(t) = f_k(\mathbf{v}_k^\top \sigma(\oplus_r \mathbf{A}_r \sigma(\oplus'_{b:r\vdash\text{depend}(k,\,b)} \mathbf{B}_r \mathbf{C}_{D_k,D_b} \mathbf{h}_b(t)))) \tag{10}$$

where $\sigma(\cdot)$ is the sigmoid function, $r$ ranges over all the rules and $r \vdash$ `depend`$(k,b)$ means "the rule $r$ proves the fact `depend`$(k,b)$". The matrices $\mathbf{A}_r \in \mathbb{R}_{\geq 0}^{D_k \times D_k}$ and $\mathbf{B}_r \in \mathbb{R}^{D_k \times D_e}$ learn how $k$ depends on each $b$. The "conversion" matrix $\mathbf{C}_{D_k,D_b} \in \mathbb{R}^{D_k \times D_b}$ projects $\mathbf{h}_b(t) \in \mathbb{R}^{D_b}$ into $\mathbb{R}^{D_k}$ for dimension compatibility with $\mathbf{B}_r$. Note that the number of $\mathbf{C}_{D,D'}$ only increases with the number of possibilities of $D_k \times D_b$ but not the number of $b$ values: the former is usually much smaller than the latter.

The aggregator $\oplus$ represents pooling operation on a set of non-negative vectors. We choose $\oplus = \sum$ and $\oplus' = \max$ because it is appropriate to sum the dependencies over all the rules but extract the "max-dependency" among all the node blocks for each rule. As shown in equation (8), the intensity of `travel(bob,chicago)` is determined by both resorts and his friends at `chicago` so these two possible motivations should be summed up. But `bob` may only stay at one of his friends' home and can only afford going to a few places, so only the "best friend" and "signature resort" matter and that is why we use max-pooling for $\oplus'$.

As a matter of implementation, we modify each `depend` rule to have the rule index $r$ as a third argument:

$$\text{depend}(k,b,r)\text{:- } \text{condition}_1,\ldots,\text{condition}_N. \tag{11}$$

This makes it possible to apply semantics-preserving transformations to the resulting Datalog program without inadvertently changing the neural architecture. Moreover, if the Datalog programmer specifies the third argument $r$ explicitly, then we do not modify that rule. As a result, it is possible for multiple rules to share the same $r$, meaning that they share parameters.

## 3.3 THE `affect` KEYWORD

We can declare database rules and facts about which events affect which node blocks using the `affect` keyword as:

$$\text{affect}(k,b)\text{:- } \text{condition}_1,\ldots,\text{condition}_N. \tag{12}$$

such that we know which node blocks to update as each event happens. For example, we can allow `travel(bob,chicago)` to update $\mathbf{h}_\text{X}(t)$ for any `X` who is a friend of `bob` and at `chicago`:

$$\text{affect}(\text{travel}(\text{bob},\text{chicago}), \text{ X}))\text{:- } \text{friend}(\text{bob},\text{X}), \text{at}(\text{X},\text{chicago}).$$

By querying the database for a given $k$ using

$$\texttt{affect}(k,\texttt{B})?$$

we get $\mathcal{B}_k^{\texttt{a}}$ that is the set of all the node blocks $b$ that instantiates $\texttt{affect}(k,b)$ where the superscript a stands for $\texttt{affect}$. Then each node block $\mathbf{h}_b(t)$ updates itself as shown in equation (2)—but it raises an important question: what are the $\mathbf{U}$ and $\mathbf{V}$ to use?

Similar to how $\mathbf{A}_r$ and $\mathbf{B}_r$ in equation (10) are declared, a $\mathbf{U}_r$ is implicitly declared by each $\texttt{affect}$ rule such that we have:

$$\boldsymbol{\psi}_{0,b,k}(t) = \oplus_{r:r\vdash\texttt{affect}(k,\,b)}\mathbf{U}_r\mathbf{w}_k \tag{13}$$

where $\oplus = \sum$. This term is analogous to the $\mathbf{U}\mathbf{w}_k$ term in section 2.1

Note that we can also modify each $\texttt{affect}$ rule (as we do for $\texttt{depend}$ in section 3.2) to have the rule index $r$ as a third argument. By explicitly specifying $r$, the Datalog programmer can allow multiple $\texttt{affect}$ rules to share $\mathbf{U}_r$.

## 3.4 THE $\texttt{update}$ KEYWORD

We can specify how node blocks update one another by using the $\texttt{update}$ keyword:

$$\texttt{update}(b',b,k)\texttt{:- condition}_1\texttt{,}\ldots\texttt{,condition}_N\texttt{.} \tag{14}$$

meaning the node block $b'$ *updates* the node block $b$ when $k$ happens. Note that $b'$ can equal $b$. It is often useful to write this rule:

$$\texttt{update(B,B,K):- affect(K,B).} \tag{15}$$

which means that whenever $\texttt{K}$ causes $\texttt{B}$ to update, $\texttt{B}$ gets to see its own previous state (as well as $\texttt{K}$).

To update the node block $b$ with event $k$, we need

$$\boldsymbol{\psi}_{1,b,k}(t) = \oplus_r \oplus'_{b':r\vdash\texttt{update}(b',\,b,\,k)}\mathbf{V}_r\mathbf{C}_{D_b,D_{b'}}\mathbf{h}_{b'}(t) \tag{16}$$

where $r$ ranges over all rules and $\oplus = \oplus' = \sum$. The matrices $\mathbf{V}_r \in \mathbb{R}^{D_b \times D_{b'}}$ learns how $b'$ updates $b$ and $\mathbf{C}_{D_b,D_{b'}} \in \mathbb{R}^{D_b \times D_{b'}}$ helps to make dimensions compatible. This term is analogous to the $\mathbf{V}\mathbf{h}(t)$ term in section 2.1.

Having equations (13) and (16), we pass $\psi_{0,b,k} + \psi_{1,b,k}$ through the activation functions and obtain the updated $\mathbf{h}_{b,\text{new}}$.

Similar to $\texttt{depend}$ and $\texttt{affect}$, we can also explicitly specify an extra argument $r$ in each $\texttt{update}$ rule to allow multiple rules to share $\mathbf{V}_r$. Parameter sharing (in $\texttt{depend}$, $\texttt{affect}$ and $\texttt{update}$) is important because it works as a form of regularization: shared parameters tend to get updated more often than the individual ones, thus leaving the latter less likely to overfit the training data when we "early-stop" the training procedure.

## 3.5 REVISITING $\texttt{is\_event}$ KEYWORD

When each event type $k$ is declared using $\texttt{is\_event}(k)$, the system automatically creates event embedding vectors $\mathbf{v}_k$ and $\mathbf{w}_k$ and they will be used in equations (10) and (13) respectively. When some event types involve many entities which results in a very large number of event types, this design might end up with too many parameters, thus being hard to generalize to unseen data.

We can allow event types to share embedding vectors by adding an extra argument to the keyword $\texttt{is\_event}$:

$$\texttt{is\_event}(k,m)\texttt{:- condition}_1\texttt{,}\ldots\texttt{,condition}_N\texttt{.} \tag{17}$$

where $m$ is an index to a pair of embedding vectors $\mathbf{v}_m$ and $\mathbf{w}_m$. There can be more than one pair that is used by an event type $k$ as shown in this example: $\texttt{is\_event(email(S,R), S)}$, $\texttt{is\_event(email(S,R), R)}$, $\texttt{is\_event(email(S,R), email)}$ and etc. Then we compute the final embedding vectors of $\texttt{email(S,R)}$ as:

$$\mathbf{v}_{\texttt{email(S,R)}} = \mathbf{v}_{\texttt{S}} + \mathbf{v}_{\texttt{R}} + \mathbf{v}_{\texttt{email}} \tag{18a}$$

$$\mathbf{w}_{\texttt{email(S,R)}} = \mathbf{w}_{\texttt{S}} + \mathbf{w}_{\texttt{R}} + \mathbf{w}_{\texttt{email}} \tag{18b}$$

Similar argument in section 3.4 applies here that sharing embedding vectors across event types is a form of regularization.

In a simplified version of our approach, we could use a homogeneous neural architecture where all events have the same dimension, etc. In our actual implementation, we allow further flexibility by using Datalog rules to define dimensionalities, activation functions, and multi-layer structures for event embeddings. This software design is easy to work with, but is orthogonal to the machine learning contribution of the paper, so we describe it in Appendix A.4.

# 4 ALGORITHMS

**Learning** Following Mei & Eisner (2017), we can learn the parameters of the proposed model by locally maximizing $\ell$ in equation (19) using any stochastic gradient method: Its log-likelihood given the sequence over the observation interval $[0, T]$ is as follows:

$$\ell = \sum\nolimits_{i:t_i \leq T} \log \lambda_{k_i}(t_i) - \int_{t=0}^{T} \sum\nolimits_{k \in \mathcal{K}} \lambda_k(t) \mathrm{d}t \tag{19}$$

The only difference is that our Datalog program affects the neural architecture, primarily by dictating that some weights in the model are structurally zero.

Concretely, to compute $\ell$ and its gradient, as each event $e_i = k_i @ t_i$ happens, we need to query the database with `depend`$(k, \mathtt{B})$? for the node blocks that each $k$ depends on in order to compute $\log \lambda_{k_i}(t_i)$ and the Monte Carlo approximation to $\int_{t=t_{i-1}}^{t_i} \sum_{k \in \mathcal{K}} \lambda_k(t) \mathrm{d}t$. Then we need to query the database with `affect`$(k, \mathtt{B})$? for the node blocks to be affected and update them. A detailed recipe is Algorithm 1 of Appendix B.1 including a down-sampling trick to handle large $\mathcal{K}$.

**Prediction** Given an event sequence prefix $k_1 @ t_1, k_2 @ t_2, \ldots, k_{i-1} @ t_{i-1}$, we may wish to predict the time and type of the next event. The time $t_i$ has density $p_i(t) = \lambda(t) \exp\left(-\int_{t_{i-1}}^{t} \lambda(s) \mathrm{d}s\right)$ where $\lambda(t) = \sum_{k \in \mathcal{K}} \lambda_k(t)$, and we choose $\int_{t_{i-1}}^{\infty} t p_i(t) \mathrm{d}t$ as the time prediction because it has the lowest expected $\mathrm{L}_2$ loss. Given the next event time $t_i$, the most likely type would simply be $\arg \max_k \lambda_k(t_i)$, but the most likely next event type *without* knowledge of $t_i$ is $\arg \max_k \int_{t_{i-1}}^{\infty} \frac{\lambda_k(t)}{\lambda(t)} p_i(t) \mathrm{d}t$. The integrals in the preceding equations can be estimated using i.i.d. samples of $t_i$ drawn from $p_i(t)$.

We draw $t_i$ using the thinning algorithm (Lewis & Shedler, 1979; Liniger, 2009; Mei & Eisner, 2017). Given $t_i$, we draw $k_i$ from the distribution where the probability of each type $k$ is proportional to $\lambda_k(t_i)$. A full sequence can be rolled out by repeatedly feeding the sampled event back into the model and then drawing the next. See Appendix B.2 for implementation details.

# 5 EXPERIMENTS

We show how to use our Datalog interface to inject inductive biases into the neural Hawkes process (NHP) on multiple synthetic and real-world datasets. On each dataset, we compare the model with modified architecture—we call it structured neural Hawkes process (or structured-NHP) with the plain vanilla NHP on multiple evaluation metrics. See Appendix C for experimental details (e.g., dataset statistics and training details). We implemented the model in PyTorch (Paszke et al., 2017).[4]

## 5.1 SYNTHETIC DATA—SUPERPOSITION OF EVENT SEQUENCES

As Mei & Eisner (2017) pointed out, it is important for a model family to handle the superposition of real-time sequences, because in various real settings, some event types tend not to interact. For example, the activities of two strangers rarely influence each other, although they are simultaneously monitored and thus form a single observed sequence.

---

[4]Code will be released upon paper acceptance.

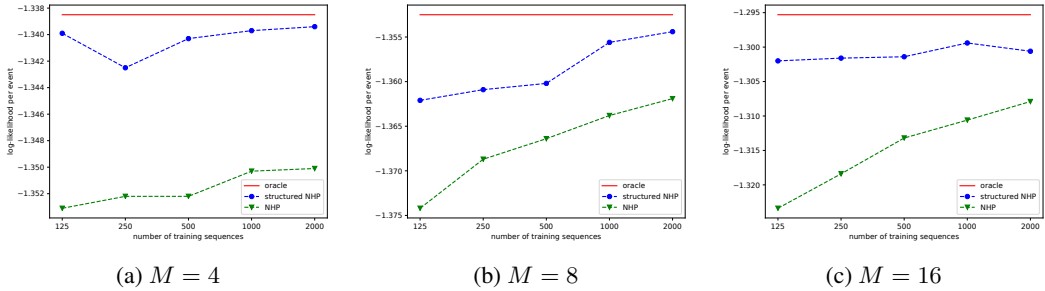

(a) $M = 4$        (b) $M = 8$        (c) $M = 16$

Figure 2: Learning curves of structured NHP and NHP on sequences drawn from the superposition of $M$ neural Hawkes processes. The red line ▬▬ indicates the *oracle* performance.

In this section, we experiment on the data *known* to be drawn from a superposition of $M$ neural Hawkes processes with randomly initialized parameters. Each process X has four event types `event(K,X)` where K can be 1, 2, 3 and 4.

To leverage the knowledge about the superposition structure, one has to either implement a mixture of neural Hawkes processes or transform a single neural Hawkes process to a superposition model by (a) zeroing out specific elements of $\mathbf{v}_k$ such that $\lambda_k(t)$ for $k \in \mathcal{K}_{\mathtt{X}}$ depends on only a subset $\mathcal{S}$ of the LSTM hidden nodes, (b) setting specific LSTM parameters such that events of type $k' \in \mathcal{K}_{\mathtt{Y}}$ don't affect the nodes in $\mathcal{S}$ and (c) making the LSTM transition matrix a blocked-structured matrix such that different node blocks don't update each other. Neither way is trivial.

With our Datalog interface, we can explicitly construct such a superposition process rather easily by writing simple datalog rules as follows:

$$\texttt{depend(event(K,X), X, X):- is\_block(X).} \tag{20a}$$
$$\texttt{affect(event(K,X), X):- is\_block(X).} \tag{20b}$$
$$\texttt{update(X, unit(X)):- is\_block(X).} \tag{20c}$$

Events of X do not influence Y at all, and processes don't share parameters.

We generated learning curves (Figure 2) by training a structured-NHP and a NHP on increasingly long prefixes of the training set. As we can see, the structured model ● substantially outperform NHP ▼ at all training sizes. The neural Hawkes process gradually improves its performance as more training sequences become available: it perhaps learns to set its $\mathbf{w}_k$ and LSTM parameters from data. However, thanks to the right inductive bias, the structured model requires much less data to achieve somewhat close to the oracle performance. Actually, as shown in Figure 2, the structured model only needs 1/16 of training data as NHP does to achieve a higher likelihood. The improvement of the structured model over NHP is statistically significant with $p$-value $< 0.01$ as shown by the pair-permutation test at all training sizes of all the datasets.

## 5.2 REAL-WORLD DATASETS

**Elevator System Dataset** (Crites & Barto, 1996). In this dataset, two elevator cars transport passengers across five floors in a building (Lewis, 1991; Bao et al., 1994; Crites & Barto, 1996). Each event type has the form `stop(C,F)` meaning that C stops at F to pick up or drop off passengers where C can be `car1` and `car2` and F can be `floor1`, ..., `floor5`. This dataset is representative of many real-world domains where individuals physically move from one place to another for, e.g., traveling, job changing, etc.

With our Datalog interface, we can explicitly express our inductive bias that each `stop(C,F)` depends on and affects the associated node blocks C and F:

$$\texttt{depend(stop(C,F), C). \quad depend(stop(C,F), F).} \tag{21a}$$
$$\texttt{affect(stop(C,F), C). \quad affect(stop(C,F), F).} \tag{21b}$$

The set of inductive biases is desirable because whether a C will head to a F and stops there is primarily determined by C's state (e.g., whether it is already on the way of sending anyone to that floor) and F's state (e.g., whether there is anyone on that floor waiting for a car).

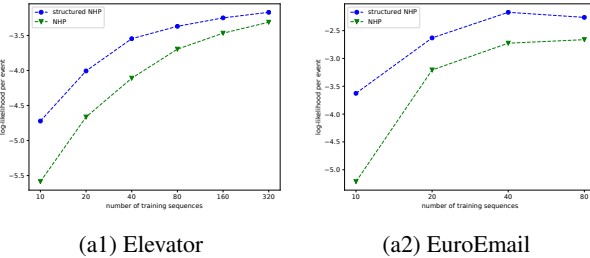

(a1) Elevator          (a2) EuroEmail

(a) Learning curves of structured NHP and NHP on the three datasets. On both datasets, the structured model substantially outperfoms NHP, especially in the data sparse scenarios.

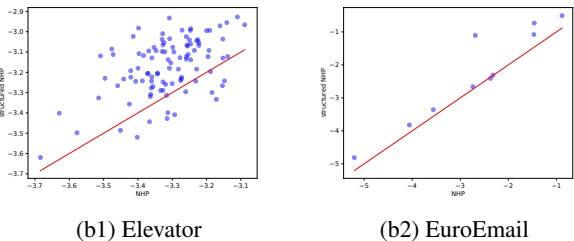

(b1) Elevator          (b2) EuroEmail

(b) Scatterplots of structured NHP vs. NHP, comparing the held-out log-likelihood of the two models (at the right end of learning curves) with respect to each test sequence. On both datasets, nearly all points lie on the top of $y = x$, since the structured model is consistently more predictive than NHP.

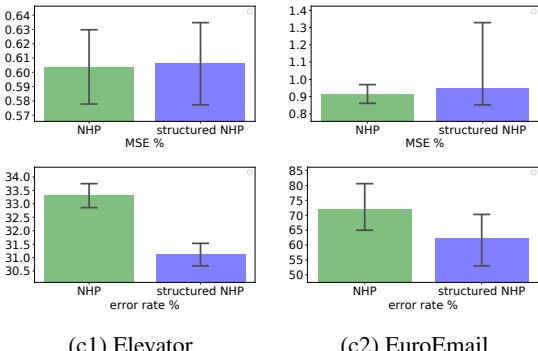

(c1) Elevator          (c2) EuroEmail

(c) Prediction results of structured NHP vs. NHP (at the right end of learning curves). Error bars show 95% bootstrap percentile confidence intervals. The MSE% denotes the mean squared error normalized by the variance of the true time interval, and error rate % denotes the fraction of the times our type prediction is incorrect.

Figure 3: Results on real-world datasets.

We also declare a global node block, `building`, that depends on and affects every event in order to compensate for any *missing* knowledge (e.g., the state of the joint controller for the elevator bank, and whether it's a busy period for the humans) and/or *missing* data (e.g., passengers arrive at certain floors and press the buttons).[5]

Appendix C.2 gives a full Datalog specification of the model that we used for the experiments in this domain. More details about this dataset (e.g. pre-processing) can be found in Appendix C.1.2.

**EuroEmail Dataset** (Paranjape et al., 2017). In this domain, we model the email communications between anonymous members of an European research institute. Each event type has the form `email(S,R)` meaning that `S` sends an email to `R` where `S` and `R` are variables that take the actual members as values.

With our Datalog interface, we can express our knowledge that each event depends on and affects its sender and receiver as the following rules:

$$\textbf{depend}(\texttt{send(S,R)}, \texttt{S}). \quad \textbf{depend}(\texttt{send(S,R)}, \texttt{R}). \tag{22a}$$

---

[5]These missing events are actually available in our domain, but we chose not to model them, to make the problem harder.

$$\mathtt{affect(send(S,R),S)}. \quad \mathtt{affect(send(S,R),R)}. \tag{22b}$$

Appendix C.2 gives a full Datalog specification of the model that we used for the experiments in this domain. More details about this dataset (e.g. pre-processing) can be found in Appendix C.1.3.

## 5.3 RESULTS

We evaluate the models in three ways as shown in Figure 3. We first plot learning curves (Figure 3a) by training a structured-NHP and an NHP on increasingly long prefixes of each training set. Then we show the per-sequence scatterplots in Figure 3b. We can see that either in learning curve or scatterplots, structured-NHP consistently outperforms NHP, which proves that structured-NHP is both more data-efficient and more predictive. Finally, we compare the models on the prediction tasks and datasets as shown in Figure 3c. We make minimum Bayes risk predictions as explained in section 4. We evaluate the type prediction with 0-1 loss, yielding an error rate. We can see, in both of Elevator and EuroEmail datasets, structured-NHP could be significantly more accurate on type prediction. We evaluate the time prediction with $L_2$ loss, and reported the mean squared error as a percentage of the variance of the true time interval (denoted as MSE%). Note that we get can MSE%=1.0 if we always predict $t_i$ as $t_{i-1} + \Delta t$ where $\Delta t$ is the average length of time intervals.

Figure 3c shows that the structured model outperforms NHP on event type prediction on both datasets, although for time prediction they perform neck to neck. We speculate that it might be because the structure information is more directly related to the event type (because of its structured term) but not time.

## 6 DISCUSSION

There has been extensive research about having inductive biases in the architecture design of a machine learning model. The epitome of this direction is perhaps the graphical models where edges between variables are usually explicitly allowed or forbidden (Koller & Friedman, 2009). There has also been work in learning such biases from data. For example, Stepleton et al. (2009) proposed to encourage the block-structured states for Hidden Markov Models (HMM) by enforcing a sparsity-inducing prior over the non-parametric Bayesian model. Duvenaud et al. (2013) and Bratières et al. (2014) attempted to learn structured kernels for Gaussian processes.

Our work is in the direction of injecting inductive biases into a neural temporal model—a class of models that is useful in various domains such as demand forecasting (Seeger et al., 2016), personalization and recommendation (Jing & Smola, 2017), event prediction (Du et al., 2016) and knowledge graph modeling (Trivedi et al., 2017). Incorporating structural knowledge in the architecture design of such a model has drawn increasing attention over the past few years. Shelton & Ciardo (2014) introduced a *factored* state space in continuous-time Markov processes. Meek (2014) and Bhattacharjya et al. (2018) proposed to consider direct dependencies among events in graphical event models. Wang et al. (2019) developed a hybrid model that decomposes exchangeable sequences into a global part that is associated with common patterns and a local part that reflects individual characteristics.

However, their approaches are all bounded to the kinds of inductive biases that are easy to specify (e.g. by hand). Our work enables people to use a Datalog program to conveniently specify the neural architecture based on a deductive database—a much richer class of knowledge than the previous work could handle. Although logic programming languages and databases have both previously been used to derive a graphical model architecture (Getoor & Taskar, 2007), we are, to the best of our knowledge, the first to develop such a general interface for a neural event model.

As future work, we hope to develop an extension where events can also trigger assertions and retractions of facts in the Datalog database. Thanks to the Datalog rules, the model architecture will dynamically change along with the facts. For example, if Yoyodyne Corp. hires Alice, then the Yoyodyne node block begins to influence Alice's actions, and $\mathcal{K}$ expands to include a new (previously impossible) event where Yoyodyne fires Alice. Moreover, propositions in the database—including those derived via other Datalog rules—can now serve as extra bits of system state that help define the $\lambda_k$ intensity functions in (1). Then the system's learned neural state $\mathbf{s}_i$ is usefully augmented by a large, exact set of boolean propositions—a division of labor between learning and expert knowledge.

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

# Appendices

## A  FRAMEWORK DETAILS

### A.1  LSTM DETAILS

In this section, we elaborate on the details of the transition function $\Psi$ that is introduced in section 2.1; more details about them may be found in Mei & Eisner (2017).

$$\mathbf{h}(t) = \mathbf{o}_i \odot (2\sigma(2\mathbf{c}(t)) - 1) \text{ for } t \in (t_{i-1}, t_i] \tag{23}$$

where the interval $(t_{i-1}, t_i]$ has consecutive observations $k_{i-1}@t_{i-1}$ and $k_i@t_i$ as endpoints. At $t_i$, the continuous-time LSTM reads $k_i@t_i$ and updates the current (decayed) hidden cells $\mathbf{c}(t)$ to new initial values $\mathbf{c}_{i+1}$, based on the current (decayed) hidden state $\mathbf{h}(t_i)$, as follows:

$$\mathbf{i}_{i+1} \leftarrow \sigma\left(\mathbf{U}_{\mathrm{i}}\mathbf{w}_{k_i} + \mathbf{V}_{\mathrm{i}}\mathbf{h}(t_i) + \mathbf{d}_{\mathrm{i}}\right) \tag{24a}$$

$$\mathbf{f}_{i+1} \leftarrow \sigma\left(\mathbf{U}_{\mathrm{f}}\mathbf{w}_{k_i} + \mathbf{V}_{\mathrm{f}}\mathbf{h}(t_i) + \mathbf{d}_{\mathrm{f}}\right) \tag{24b}$$

$$\mathbf{z}_{i+1} \leftarrow 2\sigma\left(\mathbf{U}_{\mathrm{z}}\mathbf{w}_{k_i} + \mathbf{V}_{\mathrm{z}}\mathbf{h}(t_i) + \mathbf{d}_{\mathrm{z}}\right) - 1 \tag{24c}$$

$$\mathbf{o}_{i+1} \leftarrow \sigma\left(\mathbf{U}_{\mathrm{o}}\mathbf{w}_{k_i} + \mathbf{V}_{\mathrm{o}}\mathbf{h}(t_i) + \mathbf{d}_{\mathrm{o}}\right) \tag{24d}$$

$$\mathbf{c}_{i+1} \leftarrow \mathbf{f}_{i+1} \odot \mathbf{c}(t_i) + \mathbf{i}_{i+1} \odot \mathbf{z}_{i+1} \tag{25a}$$

$$\underline{\mathbf{c}}_{i+1} \leftarrow \underline{\mathbf{f}}_{i+1} \odot \underline{\mathbf{c}}_i + \underline{\mathbf{i}}_{i+1} \odot \mathbf{z}_{i+1} \tag{25b}$$

$$\boldsymbol{\delta}_{i+1} \leftarrow f\left(\mathbf{U}_{\mathrm{d}}\mathbf{w}_{k_i} + \mathbf{V}_{\mathrm{d}}\mathbf{h}(t_i) + \mathbf{d}_{\mathrm{d}}\right) \tag{25c}$$

At time $t_i$, the updated state vector is $\mathbf{h}_{\mathrm{new}}(t_i) = \mathbf{o}_{i+1} \odot (2\sigma(2\mathbf{c}_{i+1}) - 1)$. Then, $\mathbf{c}(t)$ for $t \in (t_i, t_{i+1}]$ is given by (26), which continues to control $\mathbf{h}(t)$ except that $i$ has now increased by 1).

$$\mathbf{c}(t) \overset{\text{def}}{=} \underline{\mathbf{c}}_{i+1} + \left(\mathbf{c}_{i+1} - \underline{\mathbf{c}}_{i+1}\right) \exp\left(-\boldsymbol{\delta}_{i+1}\left(t - t_i\right)\right) \tag{26}$$

On the interval $(t_i, t_{i+1}]$, $\mathbf{c}(t)$ follows an exponential curve that begins at $\mathbf{c}_{i+1}$ (in the sense that $\lim_{t \to t_i^+} \mathbf{c}(t) = \mathbf{c}_{i+1}$) and decays, as time $t$ increases, toward $\underline{\mathbf{c}}_{i+1}$ (which it would approach as $t \to \infty$, if extrapolated).

### A.2  BOUNDARY CONDITIONS

We initialize each node block $\mathbf{h}_b(0) = \mathbf{0}$, and then have it read a special beginning-of-stream (BOS) event $\texttt{bos}@t_0$ where $\texttt{bos}$ is a special event type and $t_0$ is set to be 0. Then equations (24)–(25) define $\mathbf{c}_1$ (from $\mathbf{c}_0 \overset{\text{def}}{=} \mathbf{0}$), $\underline{\mathbf{c}}_1$, $\boldsymbol{\delta}_1$, and $\mathbf{o}_1$. This is the initial configuration of the system as it waits for the first event to happen: this initial configuration determines the hidden state $\mathbf{h}(t)$ and the intensity functions $\lambda_k(t)$ over $t \in (0, t_1]$.

The $\texttt{bos}$ event affects every node block but depends on none of them because we do not generate it. When the system is initiated, the following rule is automatically asserted by our program so users don't have to do it by themselves.

$$\texttt{affect(bos,X):- is\_block(X).} \tag{27}$$

More details about why $\texttt{bos}$ is desirable can be found in Mei & Eisner (2017).

### A.3  CONSTRUCTING THE NEURAL HAWKES PROCESS WITH OUR INTERFACE

The vanilla neural Hawkes process can be specified using our interface as follows:

$$\texttt{depend(K,global).} \tag{28a}$$

$$\texttt{affect(K,global).} \tag{28b}$$

$$\texttt{update(global,global,K).} \tag{28c}$$

where $\mathbf{h}_{\texttt{global}}(t)$ is the only node block that every event type $k$ depends on and affects. Equation (10) falls back to $f_k(\mathbf{v}_k^\top \sigma(\mathbf{A}\sigma(\mathbf{BC}\mathbf{h}_{\texttt{global}}(t))))$ which is not exactly the same with, yet at least as expressive as equation (1).

### A.4 Optional `architecture`, `input` AND `output` KEYWORDS

As discussed in section 3.5, the embedding vector of each event is just the sum of trainable vectors. Actually, we further allow users to write Datalog rules to define embedding models that have multi-layer structures and activation functions of interest.

We can define a $L$-layer neural network using the `architecture` keyword as:

$$\texttt{architecture}(n)= \texttt{transform}(\ldots(\texttt{transform}(D_0,D_1,a_1), \ldots),D_L,a_L). \tag{29}$$

where $n$ is a (structured) term as the model name, $D_0$ is the input dimension, $D_l$ and $a_l$ are the output dimension and activation type of $l$-th layer respectively. The example below defines a model named emb that has a neural layer with hyper-tangent activation followed by a linear layer.

$$\texttt{architecture}(\texttt{emb})= \texttt{transform}(\texttt{transform}(8,8,\texttt{tanh}),8,\texttt{none}) \tag{30}$$

Note that we allow using = for `architecture` to indicate that there should be only one model under each name $n$, although it is not supported in the standard datalog implementation.

We can assign to each $k$ a model $n$ and spell out its arguments $x_1, x_2, \ldots$ (to be concatenated in order) for input embedding computation using the `input` keyword:

$$\texttt{input}(k)= n(x_1, x_2, \ldots). \tag{31}$$

and follow the same format for output embedding computation using the `output` keyword. Note that we use = again. The example below means that each $\mathbf{w}_{\texttt{email(S,R)}}$ is computed by passing the concatenation of S and R into model emb and that $\mathbf{v}_{\texttt{email(S,R)}}$ is computed the same way:

$$\texttt{input}(\texttt{email}(\texttt{S},\texttt{R}))= \texttt{emb}(\texttt{S},\texttt{R}). \tag{32a}$$

$$\texttt{output}(\texttt{email}(\texttt{S},\texttt{R}))= \texttt{emb}(\texttt{S},\texttt{R}). \tag{32b}$$

## B Algorithm Details

In this section, we elaborate on the details of algorithms.

### B.1 Likelihood Computation

The log-likelihood in equation (19) can be computed by calling Algorithm 1.

The down sampling trick (line 32 of Algorithm 1) can be used when there are too many event types. It gives an unbiased estimate of the total intensity $\sum_{k\in\mathcal{K}} \lambda_k(t)$, yet remains much less computationally expensive especially when $J \ll |\mathcal{K}|$. In our experiments, we found that its variance over the entire corpus turned out small, although it may, in theory, suffer large variance. As future work, we will explore sampling from proposal distributions where the probability of choosing any $k$ is (perhaps trained to be) proportional to its actual intensity $\lambda_k(t)$, in order to further reduce the variance. But this is not within the scope of this paper.

Note that, in principle, we have to make Datalog queries after every event, to figure out which node blocks are affected by that event and to find the new intensities of all events. However, certain Datalog queries may be slow. Thus, in practice, rather than repeatedly making the same queries, we just memorize the result the first time and look it up when it is needed again.

Problems emerge when events are allowed to change the database (e.g. asserting and retracting facts as in Appendix D), then this may change the results of some queries, and thus the memos for those queries are now incorrect. In this case, we might explore using some other more flexible query language that creates memos and keeps them up to date (Filardo & Eisner, 2012).

### B.2 Thinning Algorithm for Sampling Sequences

Given an event sequence prefix $k_1@t_1, k_2@t_2, \ldots, k_{i-1}@t_{i-1}$, we can call Algorithm 2 to draw the single next event. A full sequence can be rolled out by repeatedly feeding the sampled event back into the model and then drawing the next (calling Algorithm 2 another time).

How do we construct the upper bound $\lambda^*$ (line 8 of Algorithm 2)? We express the upper bound as $\lambda^* = \sum_{k\in\mathcal{K}} \lambda_k^*$ and find $\lambda_k^* \geq \lambda_k(t)$ for each $k$. We copy the formulation of $\lambda_k(t)$ here for easy reference:

$$\lambda_k(t) = f_k(\mathbf{v}_k^\top \sigma(\oplus_r \mathbf{A}_r \sigma(\oplus'_{b\in\mathcal{B}_k^d} \mathbf{B}_r \mathbf{C}_{D_k,D_b} \mathbf{h}_b(t))))$$

---

**Algorithm 1** Log-likelihood Computation

---

**Input:** observed sequence $\mathbf{x} = k_1 @ t_1, \ldots, k_I @ t_I$ over interval $[0, T]$;
  datalog interface $d$ and model $p$;
  constant $C$, boolean flag $downsample$, down-sampling size $J$
**Output:** log-likelihood $\ell$

1: **procedure** COMPUTELOGLIKELIHOOD($\mathbf{x}, d, p, C, downsample, J$)
2:   $\ell = 0$, UPDATE($\texttt{bos}, 0, p, d$)
3:   **for** $i = 1$ **to** $I$ :                                              ▷ *loop over the sequence* $\mathbf{x}$
4:     $\lambda_{k_i}(t_i) \leftarrow$ COMPUTEINTENSITY($k_i, t_i, p, d$)        ▷ *compute intensity*
5:     $\ell \mathrel{+}= \log \lambda_{k_i}(t_i)$                              ▷ *sum up log intensity*
6:     $\Lambda \leftarrow$ COMPUTEINTEGRAL($t_{i-1}, t_i, \max\{1, \lfloor \frac{t_i - t_{i-1}}{T} CI \rfloor\}, downsample, J$)
7:     $\ell \mathrel{-}= \Lambda$
8:     UPDATE($k_i, t_i, p, d$)                                                ▷ *update node blocks*
9:   $\Lambda \leftarrow$ COMPUTEINTEGRAL($t_I, T, \max\{1, \lfloor \frac{T - t_I}{T} CI \rfloor\}, downsample, J$)
10:   $\ell \mathrel{-}= \Lambda$
11:   **return** $\ell$
12: **procedure** COMPUTEINTENSITY($k, t, p, d$)                              ▷ *compute* $\lambda_k(t)$
13:   $\mathcal{B}_k^{\mathrm{d}} \leftarrow \texttt{depend}(k, \mathbf{B})$?   ▷ *find node blocks that* $k$ *depends on*
14:   $\lambda_k(t) \leftarrow f_k(\mathbf{v}_k^\top \sigma(\oplus_r \mathbf{A}_r \sigma(\oplus'_{b \in \mathcal{B}_k^{\mathrm{d}}} \mathbf{B}_r \mathbf{C}_{D_k, D_b} \mathbf{h}_b(t))))$
15:   **return** $\lambda_k(t)$
16: **procedure** UPDATE($k, t, p, d$)                                        ▷ *update node blocks that* $k$ *affects*
17:   $\mathcal{B}_k^{\mathrm{a}} \leftarrow \texttt{affect}(k, \mathbf{B})$?  ▷ *find node blocks that* $k$ *affects*
18:   **for** $b$ **in** $\mathcal{B}_k^{\mathrm{a}}$ :
19:     query $\texttt{update}(\mathbf{B}', b, k)$?                            ▷ *find all node blocks that* $b$ *need to update its state*
20:     use equations (13) and (16) to update $\mathbf{h}_b(t)$
21: **procedure** COMPUTEINTEGRAL($t_{\mathrm{start}}, t_{\mathrm{end}}, N, downsample, J$)
22:   ▷ *compute integral over interval* $(t_{start}, t_{end}]$
23:   $\Lambda \leftarrow 0, \beta \leftarrow 1$                               ▷ *init total intensity* $\Lambda$
24:   **if** $downsample$ : $\beta \leftarrow |\mathcal{K}|/J, \mathcal{K} \leftarrow$ DOWNSAMPLE($\mathcal{K}, J$)   ▷ *down sample event types*
25:   **for** $n = 1$ **to** $N$ :
26:     draw $t \sim \mathrm{Unif}(t_{\mathrm{start}}, t_{\mathrm{end}})$
27:     **for** $k$ **in** $\mathcal{K}$ :
28:       $\lambda_k(t) \leftarrow$ COMPUTEINTENSITY($k, t, p, d$)
29:       $\Lambda \mathrel{+}= \lambda_k(t)$
30:   $\Lambda \leftarrow (t_{\mathrm{end}} - t_{\mathrm{start}}) \beta \Lambda / N$
31:   **return** $\Lambda$
32: **procedure** DOWNSAMPLE($\mathcal{K}, J$)                                 ▷ *down sample the event types*
33:   $\mathcal{K}' \leftarrow$ empty set
34:   **for** $j = 1$ **to** $J$ :
35:     uniformly draw $k' \in \mathcal{K}$, add $k'$ to $\mathcal{K}'$
36:   **return** $\mathcal{K}'$

---

---

**Algorithm 2** Thinning Algorithm for Drawing Next Event

---

**Input:** model $p$ that has read sequence $\mathbf{x} = k_1 @ t_1, \ldots, k_{i-1} @ t_{i-1}$, datalog interface $d$
**Output:** next event time $t_i$, next event type $k_i$
 1: **procedure** DRAWNEXTEVENT$(p, d)$
 2:    $t_i, \{\lambda_k(t_i)\}_{k \in \mathcal{K}} \leftarrow$ DRAWNEXTEVENTTIME$(p, d)$
 3:    draw $k_i \in \mathcal{K}$ where probability of choosing any $k$ is proportional to $\lambda_k(t_i)$
 4:    **return** $t_i, k_i$
 5: **procedure** DRAWNEXTEVENTTIME$(p, d)$
 6:     ▷ *the thinning algorithm that draws next event time by rejection sampling*
 7:    $t \leftarrow t_{i-1}$
 8:    find upper bound $\lambda^* \geq \sum_{k \in \mathcal{K}} \lambda_k(t)$ for all $t \in (t_{i-1}, \infty)$
 9:    **repeat**                                              ▷ *thinning algorithm*
10:      draw $\Delta \sim \mathrm{Exp}(\lambda^*)$, $u \sim \mathrm{Unif}(0, 1)$
11:      $t \mathrel{+}= \Delta$                  ▷ *time of next proposed event (before thinning)*
12:      compute $\lambda_k(t)$ for all $k \in \mathcal{K}$         ▷ *call the procedure line 12 of Algorithm 1*
13:    **until** $u\lambda^* \leq \sum_{k \in \mathcal{K}} \lambda_k(t)$   ▷ *thinning: accept proposed time $t$ only with prob $\frac{\sum_{k \in \mathcal{K}} \lambda_k(t)}{\lambda^*} \leq 1$*
14:    **return** $t, \{\lambda_k(t)\}_{k \in \mathcal{K}}$

---

| DATASET | $|\mathcal{K}|$ | # OF EVENT TOKENS | | | SEQUENCE LENGTH | | |
|---|---|---|---|---|---|---|---|
| | | TRAIN | DEV | TEST | MIN | MEAN | MAX |
| SYNTHETIC $M = 4$ | 16 | 42000 | 2100 | 2100 | 21 | 21 | 21 |
| SYNTHETIC $M = 8$ | 32 | 42000 | 2100 | 2100 | 21 | 21 | 21 |
| SYNTHETIC $M = 16$ | 64 | 42000 | 2100 | 2100 | 21 | 21 | 21 |
| ELEVATOR | 10 | 313043 | 31304 | 31206 | 235 | 313 | 370 |
| EUROEMAIL | 400 | 4915 | 483 | 483 | 23 | 48 | 74 |

Table 1: Statistics of each dataset.

Note that $f_k$ and $\sigma$ are positive and monotonic functions and that $\mathbf{A}_r$ are non-negative matrices. Then we can construct $\lambda_k^*$ as:

$$\lambda_k^* = f_k(\bar{\mathbf{v}}_k^\top \sigma(\oplus_r \mathbf{A}_r \sigma(\oplus'_{b \in \mathcal{B}_k^{\mathtt{d}}} \bar{\mathbf{h}}_b)))$$

where $\bar{\mathbf{v}}_k = \frac{\mathbf{v}_k + |\mathbf{v}_k|}{2}$ (i.e. only keep positive elements of $\mathbf{v}_k$) and $\bar{\mathbf{h}}_b \succeq \mathbf{B}_r \mathbf{C}_{D_k, D_b} \mathbf{h}_b(t)$ for $t \in (t_{i-1}, \infty)$.

We can construct $\bar{\mathbf{h}}_b$ by adapting the recipe in Appendix B.3 of Mei & Eisner (2017). For notation simplicity, we denote $\mathbf{B}_r \mathbf{C}_{D_k, D_b}$ as $\mathbf{G}$ and use $\mathbf{g}_d$ as its $d$-th row vector. Then the $d$-th element of $\bar{\mathbf{h}}_b$ can be expressed as $\sum_{d'=1}^{D_b} g_{dd'} h_{bd'}(t)$ where each summand $g_{dd'} h_{bd'}(t) = g_{dd'} \cdot o_{id'} \cdot (2\sigma(2c_{d'}(t)) - 1)$ is upper-bounded by $\max_{c \in \{c_{id'}, \underline{c}_{id'}\}} g_{dd'} \cdot o_{id'} \cdot (2\sigma(2c) - 1)$. Note that the coefficients $g_{dd'}$ may be either positive or negative.

## C  EXPERIMENTAL DETAILS

### C.1  DATASET STATISTICS

Table 1 shows statistics about each dataset that we use in this paper.

#### C.1.1  SYNTHETIC DATASET DETAILS

We synthesize data by sampling event sequences from different structured processes. Each structured process is a mixture model of $M$ neural Hawkes processes and each neural Hawkes `process(X)` has four event types `event1(X)`, `event2(X)`, `event3(X)` and `event4(X)`. We chose $M = 4, 8, 16$ and end up with three different datasets.

We chose the sequence length $I = 21$ and then used the thinning algorithm (Lewis & Shedler, 1979; Liniger, 2009; Mei & Eisner, 2017) to sample the first $I$ events over $[0, \infty)$. We set $T = t_I$, i.e., the time of the last generated event. We generate 2000, 100 and 100 sequences for each training, dev, and test set respectively.

### C.1.2 ELEVATOR DATASET DETAILS

We examined our method in a simulated 5-floor building with 2 elevator cars. The system was initially built in Fortran by Crites & Barto (1996) and then rebuilt in Python by Mei et al. (2019).

During a typical afternoon down-peak rush hour (when passengers go from floor-2,3,4,5 down to the lobby), elevator cars travel to each floor and pick up passengers that have (stochastically) arrived there according to a traffic profile that can be found in (Bao et al., 1994) and Mei et al. (2019).

In this dataset, each event type is `stop(C,F)` where C can be `car1` and `car2` and F can be `floor1`, ..., `floor5`. So there are 10 event types in total in this simulated building.

We repeated the (one-hour) simulation 1200 times to collect the event sequences, each of which has around 1200 time-stamped records of which car stops at which floor. We randomly sampled disjoint train, dev and test sets with 1000, 100 and 100 sequences respectively.

### C.1.3 EUROEMAIL DATASET DETAILS

EuroEmail is proposed by Paranjape et al. (2017). It was generated using email data from a large European research institute, and was highly anonymized. The emails only represent communications between institution members, which are indexed by integers, with timestamps. In the dataset are 986 users and 332334 email communications spanning over 800 days. However, most users only send or receive one or two emails, leaving this dataset extremely sparse. We extracted all the emails among the top 20 most active users, and end up with 5881 emails. We split the single long sequence into 120 sequences with average length of 48, and set the training, dev, test size as 100, 10, 10 respectively.

In this dataset, event type is defined as `send(S,R)`, where S and R are members in this organization. Then there're $20 \times 20 = 400$ different event types, where we assume that people may send emails to themselves.

### C.2 DATALOG PROGRAM DETAILS

In this section, we give a full Datalog specification of the model that we used for the experiments on each dataset.

Here is the full program for Elevator domain.

$$
\begin{align}
&\texttt{is\_block(car1).} &\text{(33a)}\\
&\texttt{is\_block(car2).} &\text{(33b)}\\
&\texttt{is\_block(floor1).} &\text{(33c)}\\
&\texttt{is\_block(floor2).} &\text{(33d)}\\
&\texttt{is\_block(floor3).} &\text{(33e)}\\
&\texttt{is\_block(floor4).} &\text{(33f)}\\
&\texttt{is\_block(floor5).} &\text{(33g)}\\
&\texttt{is\_block(building).} &\text{(33h)}\\
&\texttt{is\_car(car1).} &\text{(33i)}\\
&\texttt{is\_car(car2).} &\text{(33j)}\\
&\texttt{is\_floor(floor1).} &\text{(33k)}\\
&\texttt{is\_floor(floor2).} &\text{(33l)}\\
&\texttt{is\_floor(floor3).} &\text{(33m)}\\
&\texttt{is\_floor(floor4).} &\text{(33n)}\\
&\texttt{is\_floor(floor5).} &\text{(33o)}\\
&\texttt{is\_event(stop(C,F)):- is\_car(C),is\_floor(F).} &\text{(33p)}\\
&\texttt{depend(stop(C,F), C).} &\text{(33q)}\\
&\texttt{depend(stop(C,F), F).} &\text{(33r)}\\
&\texttt{depend(stop(C,F), building).} &\text{(33s)}
\end{align}
$$

$$\textbf{affect}(\text{stop(C,F)}, \text{C}). \tag{33t}$$
$$\textbf{affect}(\text{stop(C,F)}, \text{F}). \tag{33u}$$
$$\textbf{affect}(\text{stop(C,F)}, \text{building}). \tag{33v}$$
$$\text{update(C,C,stop(C,F))}. \tag{33w}$$
$$\text{update(F,F,stop(C,F))}. \tag{33x}$$

Here is the full program for EuroEmail domain.

$$\textbf{is\_block}(\text{user1}). \tag{34a}$$
$$\dots \tag{34b}$$
$$\textbf{is\_block}(\text{user20}). \tag{34c}$$
$$\textbf{is\_block}(\text{global}). \tag{34d}$$
$$\text{is\_user(user1)}. \tag{34e}$$
$$\dots \tag{34f}$$
$$\text{is\_user(user20)}. \tag{34g}$$
$$\textbf{is\_event}(\text{email(S,R)}) \text{:- is\_user(S),is\_user(R)}. \tag{34h}$$
$$\textbf{depend}(\text{send(S,R)}, \text{S}). \tag{34i}$$
$$\textbf{depend}(\text{send(S,R)}, \text{R}). \tag{34j}$$
$$\textbf{depend}(\text{send(S,R)}, \text{global}). \tag{34k}$$
$$\textbf{affect}(\text{send(S,R)}, \text{S}). \tag{34l}$$
$$\textbf{affect}(\text{send(S,R)}, \text{R}). \tag{34m}$$
$$\textbf{affect}(\text{send(S,R)}, \text{global}). \tag{34n}$$
$$\text{update(S,S,email(S,R))}. \tag{34o}$$
$$\text{update(R,R,email(S,R))}. \tag{34p}$$

## C.3 TRAINING DETAILS

On each dataset, given the architecture specified by the Datalog program (Appendix C.2), the hyper-parameter left to tune is the number $D$ of hidden nodes of each node block. For each model and each training size (i.e., each point in Figures 2a–2c and 3), we searched for $D$ that achieves the best performance on the dev set. Our search space is $\{4, 8, 16, 32, 64, 128\}$.

For learning, we used the Adam algorithm with its default settings (Kingma & Ba, 2015) and set minibatch size as 1. We performed early stopping based on log-likelihood on the held-out dev set.

## D ONGOING AND FUTURE WORK

We are currently exploring several extensions to deal with more complex situations.

**Interacting with the database.** Events may interact with the database by imperatively retracting and asserting certain facts. For example, when `travel(bob,newyork,chicago)` happens, the fact `in(bob,newyork)` should be retracted from the database but a new fact `in(bob,chicago)` should be inserted to it. Such retraction and assertion may be specified by using the **retract** and **assert** keywords.

**Hard constraints.** Certain datalog facts may enforce hard constraints on intensities of certain event types. For example, `in(bob,chicago)` makes `travel(bob,From,chicago)` for any `From` to have structural 0 intensities.

