# OpenReview forum: "Informed Temporal Modeling via Logical Specification of Factorial LSTMs"
_ICLR.cc/2020/Conference — Reject_

### Official Review · AnonReviewer3 · 2019-10-24
**Official Blind Review #3**

**Rating:** 3

**Review:**

This paper builds an interesting connection between Datalog rules and temporal point processes. The novelty of the approach is to factorize the latent state of LSTM into different blocks that represent three major interactions between temporal events, including: dependency, affects, and updates. The design of the node blocks within the hidden state allows the modeling of fine-grain structure of a given event type. Based on the Datalog program and the logic rules, the intensity function of the temporal point process can be formulated from facts in a database. The problem of enabling a flexible family of intensity functions is one of the most important topics in point processes, and a paper advancing knowledge in this area is certainly welcome.

The paper is in general well written. Section 2.2 can be more clarified by explicitly comparing the concepts of blocks and entities using "mind(Alice)" and "body(Alice)" before introducing the hidden state h_mind(Alice)(t). It took me some time going back and forth to understand these examples here. With respect to the design of the Datalog interface, it looks like it covers the assertion involving two arguments. Since these arguments affect the partition of the number of node blocks, it would be more clear to illustrate how to design the node blocks as the number of arguments increases (say beyond 2 arguments). In fact, if we know the number of entities in each event type, say the number of node blocks to partition is 3 per hidden state in advance, we can leverage three separate small LSTMs each of which has the private hidden state with the same number of nodes as that in one of the node blocks. Then, we can determine the interactions among these separate small LSTMs based on the logic rules, so it will be helpful to elucidate the additional advantages of partitioning these node blocks in the same hidden state. The proposed technique mainly considers how to incorporate the block design into the LSTM hidden states as a general sequence model. What is the unique characteristics of Neural Hawkes Process have been particularly exploited from this perspective? It looks like it can be applied to other LSTM-based approach as long as the predictions are functions of the hidden states. For the synthetic experiments, it is obvious that single Neural Hawkes process has more challenges to fit the mixture of processes. It will be more convincing to compare with a mixture model, like "A Dirichlet Mixture Model of Hawkes Processes for Event Sequence Clustering" with the proposed approach, and the same as in the real experiments. Also, a standard test-of-goodness fit like QQ-plot will also be more useful to improve the experiments.

**Experience Assessment:**

I have published in this field for several years.

**Review Assessment: Checking Correctness Of Derivations And Theory:**

I carefully checked the derivations and theory.

**Review Assessment: Checking Correctness Of Experiments:**

I carefully checked the experiments.

**Review Assessment: Thoroughness In Paper Reading:**

I read the paper thoroughly.

---

### Official Review · AnonReviewer2 · 2019-10-25
**Official Blind Review #2**

**Rating:** 3

**Review:**

Review for Temporal Modeling via Logical Specification of Factorial
LSTMs

This paper addresses a key problem in machine learning: how to control
the inductive bias of a model in an interpretable way.  The paper
contributes a Datalog-based language that allows a human to hand-code
structural assumptions (typically based on domain knowledge) that are
automatically translated into sparsity patterns in the parameter
matrices of an ML model (in this case, a neural Hawkes process,
although the idea would [probably] generalize to other cases).  The
language plus structured-neural-Hawkes process is demonstrated on a
few very small problems, with mixed results.

This paper is borderline.  However, I tend to favor rejection because
while the ideas are very interesting (and potentially impactful),
validation of the claims is weak.

Contributions:

On the positive side:

A Datalog interface to specifying structural zeros in parameter
matrices is a good idea.  The language is natural, and the high-level
mapping from structure and objects to low-level parameters seems
reasonable and potentially useful.

The method makes it easier to specify an inductive bias.  This is a
step in the right direction; but at its heart, this paper does not do
anything that couldn't have been done by hand - it only makes it
easier.

The method is potentially more interpretable than other attempts at
controlling inductive bias (for example, simple weight regularizaion),
but see below for why this might be a red herring.

The paper is very nicely written.  It's clear that a lot of attention
to detail went into writing it.  Well done.

Weaknesses:

There are a few major points to criticize about this paper.

First, there is no clear learning or prediction benefit.  The results
are mixed: while it appears that the SHP learns faster than the
unstructured HP, they appear to be asymptoting at the same point.
This is perhaps to be expected, as the structural zeros introduced by
the corresponding Datalog program effectively reduce the parameter
count, but the shape of the learning curves is unchanged.

(Also: please include error bars in Fig. 2(a1) and 2(a2))

The proper comparison would probably be to a low-rank parameter
matrix, where the parameter count is similarly reduced, but in an
unstructured way.  That would allow us to disentangle "parameter count
reduction" from "inductive bias", which is currently not done in the
paper.

The results in Figure 3c are mixed - it appears that SHP is only
better in 1/4 of the cases; in all other cases, the error bars seem to
indicate that there is no predictive power.

Finally, I am concerned that the method may give a false sense of
explainability to the model - why it is true that a highly structured,
symbolic language is being used to craft an inductive bias, there is
no "symbol grounding".  That is, there is no guarantee that the neural
part of the learning algorithm will use the parameters in the way the
human intended it to, because the parameters are ultimately
disconnected from the symbols.


**Experience Assessment:**

I have read many papers in this area.

**Review Assessment: Checking Correctness Of Derivations And Theory:**

I assessed the sensibility of the derivations and theory.

**Review Assessment: Checking Correctness Of Experiments:**

I carefully checked the experiments.

**Review Assessment: Thoroughness In Paper Reading:**

I read the paper at least twice and used my best judgement in assessing the paper.

---

### Official Review · AnonReviewer1 · 2019-10-30
**Official Blind Review #1**

**Rating:** 1

**Review:**

 Summary:


The paper proposed to use Datalog rules to specify the design of the LSTM architecture for event data in continuous time. The LSTM module will be used to model the rate of the events. By incorporating Datalog rules, the paper aims to encode informed inductive biases into the model.

Comments:

    After reading the entire paper, I think the main idea of this paper is to use sparse and structured weight matrices (called structural zeros” in the paper) to substitute the dense weight matrices in the original LSTM, and to split the hidden state into blocks where each block refer to a different world’s state.


         How to design the structured weight matrices and how to define the node blocks, it is informed by the  Datalog rules. This design, however, will lead to a huge weight matrix and a very  long hidden state once the types of events and number of entities grow. The proposed model will face a severe scalability issue. From this point of view, only “structural zeros” weight matrices are not enough for an elegant model.



        How to smartly share parameters and how to control the number of parameters will be an interesting direction to explore. This submission touches on this a bit but not in a principled way. For example, in Eq. 18(a) and 18(b), the embedding vectors for the grounded predicate is a summation of the embedding vectors of the entities and the predicates. This final embedding is empirically validated or is based on some permutation invariant property? This needs more clarification or some references.




        2.      The presentation needs to be polished. The current writing is not easy to follow. Especially for section 3. The architecture design needs to be clarified more. When I read this part, I felt a little difficult to map the Datalog rules to your model.
        3        Since you are learning the vector embeddings for event types and entities, what are the advantages of this compared to the marked point process model, where the event types and entities are treated as discrete markers and are a much more parsimonious model. The Datalog rules can also be defined on the marker level by introducing a structured dependency structure over the markers. What are the potential benefits of learning the embeddings? The explanation is missing in this paper.

       4     Lack of references. The proposed neural-symbolic architecture shares some similarities to the following papers:
          (1) End-to-End Differentiable Proving

          (2) DeepProbLog: Neural Probabilistic Logic Programming

          (3) Neural Logic Machines.

        What are your contributions and differences in terms of the neural-symbolic architecture design?


       As for introducing logic rules to guide event predication, this is not a new topic. Here is a list of references:


         (1) PEL-CNF: Probabilistic event logic conjunctive normal form for video interpretation.
          (2) A general framework for recognizing complex events in Markov logic.

          (3) Learning Bayesian networks for clinical time series analysis.

          (4) Logical Hierarchical Hidden Markov Models for Modeling User Activities.

          (5) Slice Normalized Dynamic Markov Logic Networks.
       5.         Lack of strong baselines. The paper only did a small-scale experiment study. It only compares a neural Hawkes process model. The experimental evaluation also needs stronger baselines. Specifically, methods that can handle continuous-time (e.g. marked point process) or probabilistic logic methods that can discretize time (as mentioned in the above references). The baselines are not quite strong and appear a bit arbitrary in the paper.


**Experience Assessment:**

I have published in this field for several years.

**Review Assessment: Checking Correctness Of Derivations And Theory:**

I assessed the sensibility of the derivations and theory.

**Review Assessment: Checking Correctness Of Experiments:**

I assessed the sensibility of the experiments.

**Review Assessment: Thoroughness In Paper Reading:**

I read the paper at least twice and used my best judgement in assessing the paper.

---

### Author Response · Authors · 2019-11-14
**Thanks and clarification.**

Thanks very much to the reviewers -- these are high-quality reviews.  We appreciate the time you spent on the paper and the thoughtful feedback.

Our presentation was written too quickly, and more careful writing would have answered some of your main concerns.  In the model, we do have ways to handle parameter sharing (last sentences in sections 3.2, 3.3 and 3.4) and event type composition (section A.4). In the experiments, we tuned hyper-params (including # hidden nodes) for the baseline model (which is indeed a strong multivariate point process) as well as for our model, so the gain is from the design improvement.  We will clarify these points in the next version.

Unrelatedly, our technical approach has evolved and deepened considerably since we submitted this version. We don't think it would be appropriate to deeply change our ICLR submission at this late stage, so we'll just submit our next version to the next conference. We'll certainly take your comments into account as well -- thanks again!

---

### Decision · Program_Chairs · 2019-12-19

**Decision:**

Reject

**Comment:**

While reviewers find this paper interesting, they raised number of concerns including the novelty, writing, experiments, references and clear mention of the benefit. Unfortunately, excellent questions and insightful comments left by reviewers are gone without authors’ answers.